# Flow Velocity Field Measurement of Vertical Upward Oil–Water Two-Phase Immiscible Flow Using the Improved DPIV Algorithm Based on ICP and MLS

**Lianfu Han, Yao Cong, Xingbin Liu * and Changfeng Fu ***

College of Electronics Science, Northeast Petroleum University, Daqing 163318, China

* Correspondence: dlts_liuxb@petrochina.com.cn (X.L.); changfengfunepu@163.com (C.F.);
  Tel.: +86-198-459-25779 (X.L.); +86-182-496-29368 (C.F.)

**Abstract:** Flow velocity field measurement is important for analyzing flow characteristics of oil–water two-phase immiscible flow in vertical well. Digital particle image velocimetry (DPIV) is an effective velocity field measurement method that has overcome single point measurement limitation of traditional instruments. However, multiphase flow velocity fields generated by DPIV are often accompanied by local false vectors caused by image mismatching, which leads to measurement results with low accuracy. In this paper, the reasons for oil–water two-phase immiscible flow image mismatching in inner diameter 125 mm vertical pipe is identified by studying the DPIV calculation process. This is mainly caused by image noise and poor window following performance that results from poor deformation performance of the interrogation window. To improve deformation performance of the interrogation window, and thus improve the accuracy of the algorithm, iterative closest point (ICP) and moving least squares (MLS) are introduced into the window deformation iterative multigrid algorithm in DPIV postprocessing algorithm. The simulation showed that the improved DPIV algorithm had good matching performance, and thus the false vector was reduced. The experimental results showed that, in light of the present investigation, on average, the improved DPIV algorithm is found to yield an accuracy improvement of ~6%; the measurement uncertainty and reproducibility of the improved DPIV algorithm were $0.149 \times 10^{-3}$ m/s and 1.98%, respectively.

**Keywords:** oil–water two-phase immiscible flow; vertical pipe; digital particle image velocimetry; window deformation iterative multigrid; iterative closest point; moving least squares

## 1. Introduction

Flow velocity field measurement is important for analyzing the flow characteristics of oil–water two-phase immiscible flow, which directly affects the interpretation of logging data and design of logging instruments [1,2]. Flow velocity field measurement methods include the capacitance method [3], conductance method [4], and electromagnetic method [5]. Current oilfield flow velocity field measurement is more inclined toward velocity distribution visualization measurement, which can obtain more detailed flow velocity field information and observe the flow characteristics more easily to guide oilfield development [6]. It is very difficult for the traditional measurement method to meet this requirement. Therefore, a new flow velocity field measurement method must be adopted. Particle image velocimetry (PIV) is an effective flow velocity field measurement technology in hydromechanics [7,8]. Its characteristics of nondisturbance, noncontact, and full-field measurement [9] meet the requirements of oilfields. Therefore, it has promising potential applications.

PIV was originally developed from laser speckle velocimetry [10]. Then, PIV was gradually digitized into digital particle image velocimetry (DPIV) with the advent of digital technology [11–13]

and different illumination solutions, like LED or flash lamps were proposed for specific applications, where a laser source is inadequate due to the high tracers density (e.g., multiphase flows and liquid-granular flows). The intrinsic principle of DPIV is to capture the pattern movements by means of a cross-correlation optimization, as shown in Figure 1a [14]. During the process of measuring the flow velocity field, an appropriate amount of tracer is evenly sprinkled into the flow body. Then, a computer synchronously controls the pulsed laser light source and high-speed camera to illuminate the flow velocity field and record the instantaneous flow state [15–17]. Some scholars carry out hardware improvement at the foundation of the classical PIV system: Sarno [18] uses the LED lamp placed in front of the pipe to measure the velocity fields of granular flows and Estevadeordal et al. [19], Hessenkemper et al. [20], and Cerqueira et al. [21] use a light-emitting diode (LED) backlight source to replace the laser source, which has also achieved good measurement performance, as shown in Figure 1b. Finally, the recorded digital image is transmitted to the computer for further processing to obtain the instantaneous flow velocity field [22].

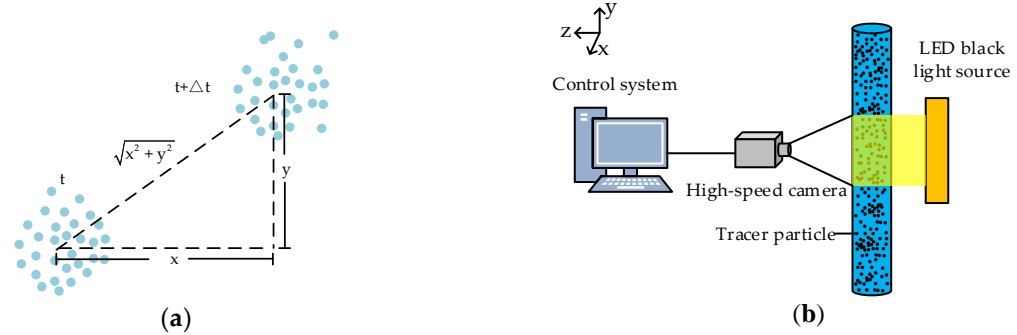

**Figure 1.** Principle of digital particle image velocimetry (DPIV) technology: (**a**) the velocity calculation method and (**b**) DPIV system.

The traditional DPIV postprocessing algorithm is based on cross-correlation and the fast Fourier transform (FFT) algorithm [23,24]. Although simple, it has many limitations. In Figure 2, the yellow points stand for the particles whose velocity is very high, the blue points stand for the particles those are still in the interrogation window, the brown points stand for the new particles, and the red points stand for the particles, which are out of the interrogation window for the displacement increases. As shown in Figure 2a, the consistent interrogation window size and position of the two images remains unchanged, so the red points run out the interrogation window. The matching particles in the interrogation window decrease as the displacement increases as a result of the consistent interrogation window size and position of the two images [25]. The accuracy of DPIV is affected by the numbers of matching particles, so this make the decrease in accuracy of DPIV. By offsetting the interrogation window using the predetermined pixel displacement to maximize the particle matching rate, Westerweel et al. [26] and Gui et al. [27] proposed discrete and continuous window offset technologies, respectively. However, this still has limitations in a flow field with a velocity gradient. As shown in Figure 2b, the yellow points still run out the interrogation. As shown in Figure 2c, Huang et al. [28] proposed the concept of image deformation, in which the interrogation window is deformed according to the kinematics formula to improve the calculation accuracy in a complex flow state. Scarano and Riethmuller [29] adopted a Taylor expansion to fit the displacement distribution in the deformation algorithm and combined it with an iterative multigrid algorithm to create the window deformation iterative multigrid (WIDIM) algorithm, which is now widely accepted. To reduce the amount of calculation, Scarano [30] used bilinear interpolation to construct the pixel point displacement field for image deformation. Astarita [31] analyzed the bilinear, shift bilinear, B-spline, and FFT interpolation methods. The B-spline method has the highest interpolation accuracy, and the second-order B-spline can be applied to most actual scenarios.

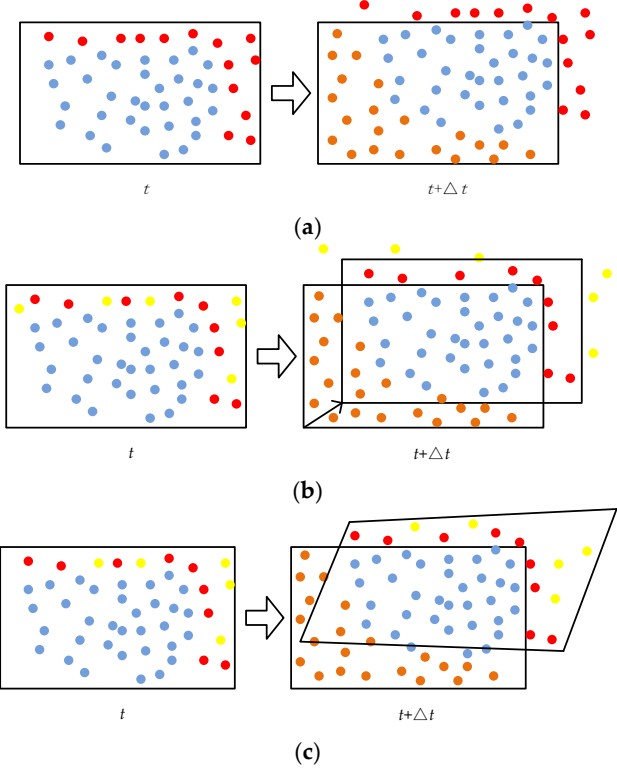

**Figure 2.** Particle motion in interrogation area: (**a**) particles run out the interrogation window, (**b**) particle motion with large velocity, and (**c**) window deformation.

There are still many limitations in multiphase flow measurement by DPIV, such as the shielding of oil against tracer particles in oil–water two-phase immiscible flow [32]. Xu et al. [33] applied DPIV to the small diameter horizontal oil–water two-phase immiscible flow with oil droplets as tracer particles, and the results represented the average accumulation flow velocity of oil droplets in the shooting direction. Kong et al. [34] applied DPIV to oil–water two-phase immiscible flow in a small diameter horizontal well, and modified the window overlap method in the WIDIM algorithm to improve the measurement accuracy. Although these algorithms are effective in their application filed, they do not perform well for oil–water two-phase immiscible flow measurement in an inner diameter 125 mm vertical upward pipe. The inner diameter 125 mm vertical well is the most widely used in oilfields. At low flow velocity and high water cut (the water volume percentage), water is the continuous phase and oil is the dispersed phase and the fluid appears as the bubbly flow state in the well. Oil droplets are small in inner diameter 125 mm vertical upward pipe, with good follow ability, and could be used as tracer particles to obtain the shooting direction cumulative average flow velocity distribution information in the plane.

At present, the following problems exist in the application of the WIDIM algorithm in DPIV to oil–water two-phase immiscible flow velocity field measurement in an inner diameter 125 mm vertical pipe. (1) As a result of the large diameter, the relative motion range of oil droplets in the interrogation window is very large. Therefore, the following performance of the interrogation window is required to be good enough. However, the following performance of the interrogation window does not meet the demand of the velocity measurement of 125 mm vertical upward oil–water two-phase immiscible flow. (2) Compared to small-scale laboratory investigations, the larger spatial domain increases the illumination issues and oil droplets shielding problems, especially if the LED source is placed behind the pipe. (3) Because the application object of DPIV is two-phase flow with oil tracers, the gray image of two-phase flow is more complex than that of single-phase flow. This makes the cross-correlation calculation prone to mismatching and generating plenty of false vectors so as to leads to the measurement error. The error may be amplified during the iteration process of the algorithm,

especially the initial calculation error, thereby seriously affecting image deformation performance. Hence, the iteration calculation accuracy should be improved as far as possible; thus, the method how to increase the accuracy of the iteration calculation should be studied. (4) There are missing data at the displacement field boundary in the interpolation process of WIDIM algorithm, which also affect deformation performance. Thefore, the displacement field boundary problem should be studied.

In this paper, the iterative closest point (ICP) algorithm is introduced into the WIDIM algorithm to calculate the initial displacement field to improve the accuracy of the initial iteration. Moreover, the moving least squares (MLS) algorithm is used to fit the displacement field to obtain the fitting value of the boundary displacement in the interpolation process. The improved DPIV algorithm is applied to oil–water two-phase immiscible flow measurement in an inner diameter 125 mm vertical upward pipe, and the flow velocity distribution characteristics in plane are studied.

## 2. Improvement of DPIV Algorithm

### 2.1. Window Deformation Iterative Multigrid Algorithm

The WIDIM algorithm is an effective DPIV postprocessing algorithm. The WIDIM algorithm process is shown in Figure 3 [35]. Firstly, a large interrogation window is used for the initial iteration calculation to obtain the initial displacement field. The displacement field is interpolated to obtain the pixel displacement field. Secondly, the interrogation window is deformed according to the pixel displacement field, and the interrogation window is reduced. The second iteration is performed and the displacement field is updated. Finally, the calculation and update are performed continuously until the interrogation window is reduced to the specified size.

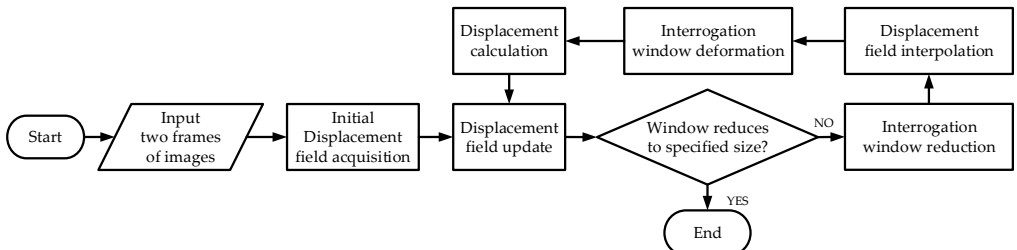

**Figure 3.** Window deformation iterative multigrid (WIDIM) algorithm flow chart.

### 2.2. Improved WIDIM Algorithm

#### 2.2.1. ICP Algorithm

In the WIDIM algorithm, typically, mismatched pixels exist in the large interrogation window in the initial iteration. Moreover, the vertical upward oil–water two-phase immiscible flow DPIV image contains more abundant gray information compared with single-phase flow images, including noise that results from uneven illumination. Hence, the displacement calculated by cross-correlation in the iteration tends to produce a large number of errors, especially in the initial iteration. These errors may be amplified many times in the subsequent iteration process, thereby affecting the deformation of the interrogation window and resulting in the reduction of the image matching rate. Therefore, it is necessary to take measures to improve the accuracy of the displacement field in the iteration.

ICP is a type of point cloud data registration method that can achieve the accurate registration of large amounts of data [36]. The three-dimensional (3D) ICP is applied to 2D DPIV image registration simply by projecting the 2D gray image into 3D space, as shown in Figure 4. Compared with cross-correlation matching, ICP is less affected by noise [37]. Moreover, the plane rotation of oil droplets is taken into account in the registration process. Table 1 shows the comparison between ICP and cross-correlation algorithm applied on the 117 pixel × 179 pixel image.

**Table 1.** The comparison between ICP and normalized cross-correlation.

| Algorithm | Run-Time | Registration Precision |
|:---:|:---:|:---:|
| ICP | 0.3315 s | 90.3% |
| Cross-correlation | 0.1010 s | 84.2% |

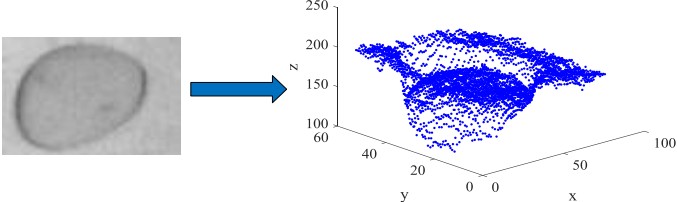

**Figure 4.** A three-dimensional projection of a two-dimensional grayscale image.

The registration precision means the accuracy degree of the position of the first image's object in the second image obtained by algorithms, and that of ICP is higher than that of cross-correlation. Although its run-time is longer than cross-correlation's run-time due to the complicacy and expensive computation of ICP, we still use ICP in the subsequent PIV passes. Because the proposed approach is not meant to be used for real-time measurements. The ICP algorithm searches for the optimal transformation $R$, $T$ according to the nearest neighbor principle and minimizes the objective function repeatedly, thus conducting spatial matching for the two point sets. Given target point set $p$, where the coordinate is $\{p_i | p_i \in R^3, i = 1, 2, \ldots, N_p\}$. Given reference point set $q$, where the coordinate is $\{q_i | q_i \in R^3, i = 1, 2, \ldots, N_q\}$. The transform and target function [38], respectively, are

$$q'_i = Rp_i + T, \tag{1}$$

$$\theta = \sum_{i=1}^{N_p} \|q_i - q'_i\|^2, \tag{2}$$

where, $R$ and $T$ are the rotation matrix and translation vector, respectively.

Singular value decomposition (SVD) is adopted to solve rotation matrix $R$ and translation vector $T$, and the point set $p$ and $q$ are transformed as follows

$$P_i = p_i - \frac{1}{N_p} \sum_{j}^{N_p} p_j, \tag{3}$$

$$Q_i = q_i - \frac{1}{N_p} \sum_{j=1}^{N_p} q_j. \tag{4}$$

Then

$$H = \sum_{i=1}^{N_p} Q_i P_i^{\mathrm{T}}. \tag{5}$$

SVD is performed for $H$:

$$H = \Phi \Lambda \Psi^{\mathrm{T}}, \tag{6}$$

where $\Phi$ and $\Psi$ are all unitary matrixes that contain left singular vector and right singular vector, respectively; $\Lambda$ is the singular matrix.

Then the calculation formula of $R$ and $T$ are as follows

$$R = \Phi \Psi^{\mathrm{T}}, \tag{7}$$

$$T = \frac{1}{N_p} \sum_i^{N_P} \boldsymbol{q}_i - \boldsymbol{\Phi}\boldsymbol{\Psi}^{\mathrm{T}} \frac{1}{N_p} \sum_{i=1}^{N_P} \boldsymbol{p}_i. \tag{8}$$

The target point set, $p$, is updated according to Equation (1); continue to solve new $\boldsymbol{R}$, $\boldsymbol{T}$. Updating and solving $\boldsymbol{R}$, $\boldsymbol{T}$ are performed iteratively until $\theta$ is less than the set threshold.

### 2.2.2. MLS Algorithm

In the DPIV window deformation algorithm, a deficiency in the boundary node value often exists in the displacement field to be interpolated, which is caused by assigning the obtained displacement value by default to the grid center node. As shown in Figure 5, the absence of the value at the field boundary leads to a number of "NaN" values in the interpolation result.

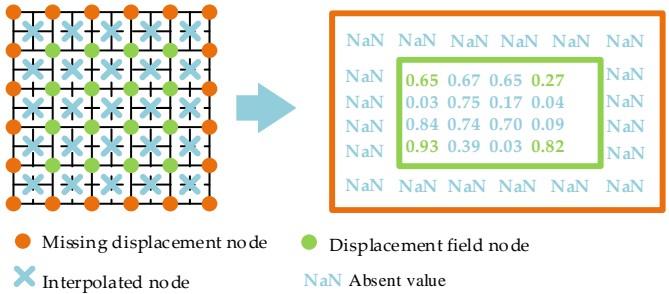

**Figure 5.** Displacement field with boundary node value deficiency and interpolation results.

The interrogation window is deformed according to the interpolation results. The interpolation results is close to actual displacement distribution so that the deformed interrogation window has good following performance. However, the "NaN" values seriously affect the deformation effect of the interrogation window. The key to solve this problem is to assign values to the displacement deficient. The assigned value is obtained by considering all the displacement field information. Therefore, the WIDIM algorithm is improved by the MLS algorithm. The current displacement field is fitted by MLS before each interpolation, and the deficient boundary values are supplemented by surface fitting functions. The MLS algorithm is an enhanced version of the least square method algorithm, and can solve fitting smoothness and localization problems. As shown in Figure 5, the displacement field is expressed as $D(x_{ij}, y_{ij}, z_{ij})$ ($i = 1, \dots, 6, j = 1, \dots, 6$). $x_{ij}$ and $y_{ij}$ are the coordinate values of node ($i, j$), and $z_{ij}$ is the corresponding displacement value. Therefore, the surface fitting process is as follows:

The surface fitting function [39] of the displacement field in Figure 5 can be expressed as

$$f(x, y) = \sum_k^l \alpha_k(x, y)\mu_k(x, y) = \boldsymbol{\alpha}(x, y)\boldsymbol{\mu}^{\mathrm{T}}(x, y), \tag{9}$$

where, $\boldsymbol{\mu}(x, y)$ is basis function and has multiple forms that can be written as

$$\begin{aligned} \boldsymbol{\mu}(x, y) \quad &= [\mu_1(x, y), \mu_2(x, y), \dots, \mu_l(x, y)] \\ &= \begin{cases} [1, x, y], & l = 3, \\ [1, x, y, x^2, xy, y^2], & l = 6, \\ [1, x, y, x^2, xy, y^2, x^3, x^2y, xy^2, y^3], & l = 10. \end{cases} \end{aligned} \tag{10}$$

To meet the requirements of both accuracy and computational complexity, the value of $l$ in this paper is 6.

The local approximation function of Equation (9) in the domain of $(x, y)$ is defined as

$$f[(x,y),(\overline{x},\overline{y})] = \sum_{k=1}^{l} \alpha_k(x,y)\mu_k(\overline{x},\overline{y}) = \boldsymbol{\alpha}(x,y)\boldsymbol{\mu}^{\mathrm{T}}(\overline{x},\overline{y}), \tag{11}$$

where, $\alpha_k$ $(x, y)$ is obtained by the weighted least squares fitting of local approximation function $f[(x,y),(\overline{x},\overline{y})]$. $\alpha_k$ $(x, y)$ should make $f[(x,y),(\overline{x},\overline{y})]$ as close to the theory value as possible. For the displacement field in Figure 5, this means minimizing

$$\begin{aligned}
I &= \sum_{i=2}^{5}\sum_{j=2}^{5} \omega\left(\sqrt{\left(x-x_{ij}\right)^2 + \left(y-y_{ij}\right)^2}\right)\left[f\left((x,y),\left(x_{ij},y_{ij}\right)\right) - z_{ij}\right]^2 \\
&= \sum_{i=2}^{5}\sum_{j=2}^{5} \omega\left(\sqrt{\left(x-x_{ij}\right)^2 + \left(y-y_{ij}\right)^2}\right)\left[\sum_{k}^{l}\alpha_k(x,y)\mu_k\left(x_{ij},y_{ij}\right) - z_{ij}\right]^2,
\end{aligned} \tag{12}$$

where, $\omega\left(\sqrt{\left(x-x_{ij}\right)^2 + \left(y-y_{ij}\right)^2}\right)$ is the weight function with the property of a compact support set and $(x_{ij}, y_{ij})$ is the coordinate of node $(i, j)$ in the displacement field in addition to the point in the compact support domain. Equation (11) is written as

$$I = (\boldsymbol{\alpha}\boldsymbol{M} - \boldsymbol{Z})\boldsymbol{\Omega}(\boldsymbol{\alpha}\boldsymbol{M} - \boldsymbol{Z})^{\mathrm{T}}, \tag{13}$$

where, $\boldsymbol{Z} = [z_{22}, z_{32}, z_{42}, \ldots, z_{ij}]$, $\boldsymbol{M} = [\boldsymbol{\mu}^{\mathrm{T}}$ $(x_{22}, y_{22})$, $\boldsymbol{\mu}^{\mathrm{T}}$ $(x_{32}, y_{32})$, $\ldots$, $\boldsymbol{\mu}^{\mathrm{T}}$ $(x_{ij}, y_{ij})]$, and $\boldsymbol{\Omega} = \mathrm{diag}\ (\omega)$, $i, j = 2, \ldots, 5$.

Consider the extremum of $I$, that is,

$$\frac{\partial I}{\partial \alpha} = \boldsymbol{\Gamma}(x,y)\boldsymbol{\alpha}^{\mathrm{T}}(x,y) - \boldsymbol{E}(x,y)\boldsymbol{Z} = 0, \tag{14}$$

where, $\boldsymbol{\Gamma}$ $(x, y) = \boldsymbol{M}^{\mathrm{T}}\boldsymbol{\Omega}\boldsymbol{M}$, $\boldsymbol{E}$ $(x, y) = \boldsymbol{M}^{\mathrm{T}}\boldsymbol{\Omega}$.

And

$$\boldsymbol{\alpha}(x,y) = \boldsymbol{Z}\left(\boldsymbol{\Gamma}^{-1}(x,y)\boldsymbol{E}(x,y)\right)^{\mathrm{T}}. \tag{15}$$

Substitute Equation (15) into Equation (11) to obtain the equation of the fitted surface

$$f(x,y) = \boldsymbol{Z}\left(\boldsymbol{\Gamma}^{-1}(x,y)\boldsymbol{E}(x,y)\right)^{\mathrm{T}}\boldsymbol{\mu}(x,y)^{\mathrm{T}}. \tag{16}$$

Then the displacement field in Figure 5 can be expressed as

$$F_{ij} = \begin{cases} f\left(x_{ij}, y_{ij}\right), & i = 1, 6 \text{ or } j = 1, 6, \\ z_{ij}, & else. \end{cases} \tag{17}$$

### 2.2.3. Improved DPIV Algorithm Based on ICP and MLS

The specific steps of the improved DPIV algorithm based on ICP and MLS are as follows.

Step 1: Images $G_1$ and $G_2$ are divided into multiple sub-images, consider the large interrogation window to conduct initial registrations of these sub-images. The initial displacement field $\boldsymbol{U}, \boldsymbol{V}$ is calculated according to Equations (1)–(8):

$$\boldsymbol{U} = \begin{bmatrix} T_{11}(1) & T_{12}(1) & \cdots & T_{1j}(1) \\ T_{21}(1) & T_{22}(1) & \cdots & T_{2j}(1) \\ \vdots & \vdots & \ddots & \vdots \\ T_{i1}(1) & T_{i2}(1) & \cdots & T_{ij}(1) \end{bmatrix}, \boldsymbol{V} = \begin{bmatrix} T_{11}(2) & T_{12}(2) & \cdots & T_{1j}(2) \\ T_{21}(2) & T_{22}(2) & \cdots & T_{2j}(2) \\ \vdots & \vdots & \ddots & \vdots \\ T_{i1}(2) & T_{i2}(2) & \cdots & T_{ij}(2) \end{bmatrix}, \tag{18}$$

where, $U$ and $V$ are horizontal and vertical displacement matrices, respectively. $T_{ij}$ is the translation vector of each sub-image calculated by ICP.

Step 2: Reduce the interrogation window size, perform MLS fitting for the initial displacement field. Boundary fitting value of the displacement field is obtained according to Equations (9)–(12).

Step 3: Construct pixel displacement field $U'$, $V'$ using bicubic uniform B-spline surface interpolation. Calculate the bicubic uniform B-spline surface slice as follows

$$\Theta(\xi, \zeta) = \lambda B_q C B_w^{\mathrm{T}} \gamma^{\mathrm{T}}, \tag{19}$$

where, $\lambda = [\xi^3, \xi^2, \xi, 1]$, $\gamma = [\zeta^3, \zeta^2, \zeta, 1]$ and $B_q$, $B_w$ represent the quad node B-spline basis function that corresponds to control vertex $C$. When $\xi$, $\zeta$ traverse between [0, 1], the displacement value at any point on surface slice $\Theta$ can be obtained.

Step 4: Deform the image according to pixel displacement field $U'$, $V'$:

$$G'_2(X, Y) = G_2(X + U', Y + V'); \tag{20}$$

Step 5: Perform ICP on $G_1$, $G'_2$ with a small interrogation window to obtain the new displacement field $U^e$, $V^e$:

Step 6: The displacement field is updated:

$$U' = U' + U^e, V' = V' + V^e; \tag{21}$$

Step 7: Iteratively perform Steps 1–6 until the interrogation window is reduced to the specified size, and the final flow velocity field is calculated as

$$L = \frac{\sqrt{U'^2 + V'^2}}{\Delta t}, \tag{22}$$

where, $\Delta t$ is the time interval.

For the convenience of illustration, the DPIV algorithm based on WIDIM algorithm is called the WIDIM-DPIV algorithm, and the improved WIDIM-DPIV algorithm based on ICP and MLS is called the IM-WIDIM-DPIV algorithm.

## 3. Verification of the IM-WIDIM-DPIV Algorithm

To verify the feasibility of the IM-WIDIM-DPIV algorithm, the DPIV combination flow image which was synthesized using MATLAB. The known speed of local oil–water two-phase immiscible flow grayscale image is 1000 pixel/s, and the interval is 0.001 s. The results are shown in Figure 6.

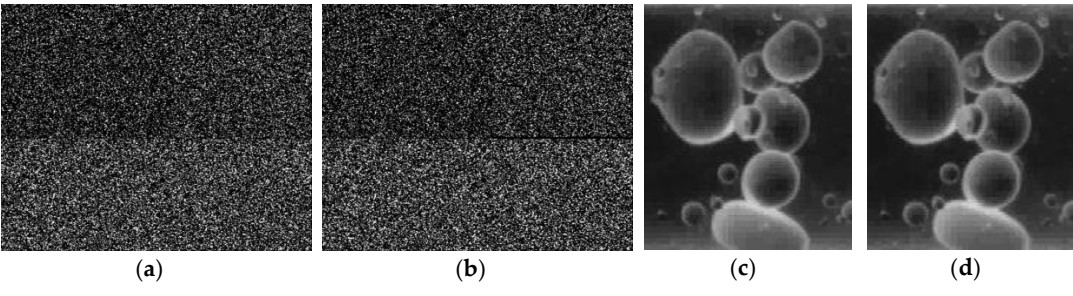

(a)　　　　　　　　(b)　　　　　　　　(c)　　　　　　　　(d)

**Figure 6.** *Cont.*

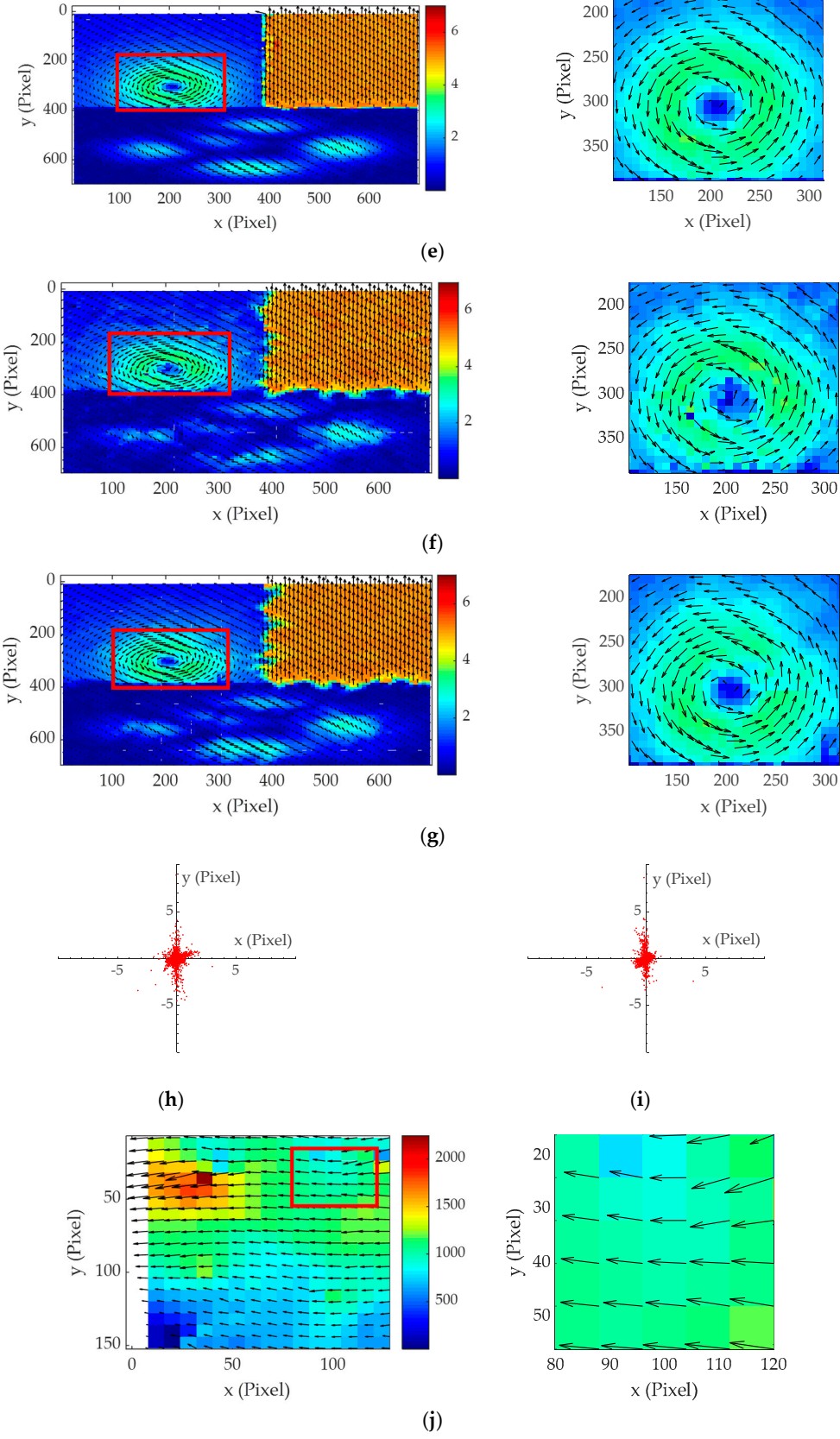

**Figure 6.** *Cont.*

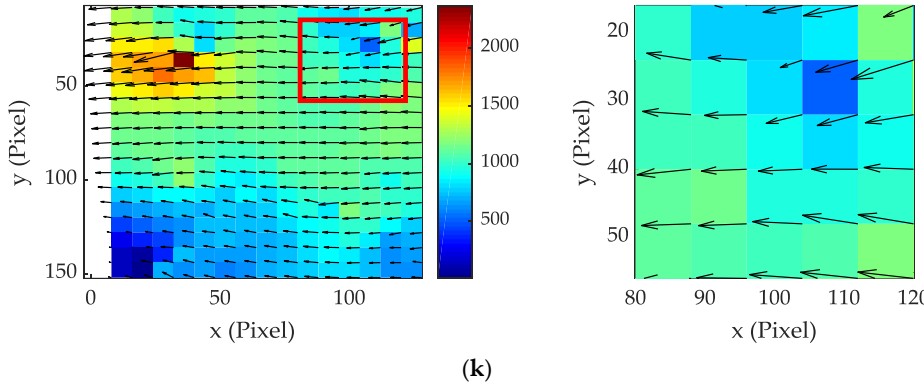

(**k**)

**Figure 6.** IM-WIDIM-DPIV algorithm verification. (**a**) The synthesized image generated by MATLAB. (**b**) The image produced by moving the particles in Figure 6a according to a given displacement field. (**c**,**d**) The grayscale image pair of local oil–water two-phase immiscible flow. (**e**) The given displacement field in Figure 6a,b. (**f**) The displacement field of Figure 6a,b calculated by the WIDIM-DPIV algorithm. (**g**) The displacement field of Figure 6a,b calculated by the IM-WIDIM-DPIV algorithm. (**h**) The WIDIM-DPIV algorithm error plot of synthesized image. (**i**) The IM-WIDIM-DPIV algorithm error plot of synthesized image. (**j**) The flow velocity field of Figure 6c,d calculated by the WIDIM-DPIV algorithm. (**k**) The flow velocity field of Figure 6c,d calculated by the IM-WIDIM-DPIV algorithm.

For the synthesized DPIV images, the flow velocity field obtained by the IM-WIDIM-DPIV algorithm was closer to the theory flow velocity field, and some of false vectors in Figure 6f were revised by the IM-WIDIM-DPIV algorithm. In Figure 6h,i, the horizontal coordinate represents the error of the two algorithms in the x direction, and the vertical coordinate represents the error of the two algorithms in the y direction. Points in Figure 6i are closer to the origin of coordinate than points in Figure 6h, indicating that the error of IM-WIDIM-DPIV is smaller than that of WIDIM-DPIV. The average absolute error of the WIDIM-DPIV algorithm was 0.2916 pixels, and that of the IM-WIDIM-DPIV algorithm was 0.1628 pixels. For grayscale images, the average velocity calculated by the WIDIM-DPIV was 928.0 pixels, and the average velocity calculated by the IM-WIDIM-DPIV algorithm was 1034.8 pixels, which improved the velocity measurement accuracy by 3.72%. The run-time and measurement accuracy of two algorithms are shown in Table 2. The measurement accuracy of IM-WIDIM-DPIV algorithm is higher than that of WIDIM-DPIV algorithm, while the run-time of IM-WIDIM-DPIV algorithm is longer than that of WIDIM-DPIV algorithm.

**Table 2.** The run-time and measurement accuracy of WIDIM-DPIV algorithm and IM-WIDIM-DPIV algorithm.

| Algorithm | Synthesized Image | | Grayscale Image | |
|---|---|---|---|---|
| | Run-Time | Measurement Accuracy | Run-Time | Measurement Accuracy |
| WIDIM-DPIV | 51.25 s | 86.7% | 2.34 s | 92.8% |
| IM-WIDIM-DPIV | 143.27 s | 93.8% | 8.27 s | 96.5% |

According to the above analysis, the measurement accuracy of IM-WIDIM-DPIV algorithm is better than the measurement accuracy of WIDIM-DPIV algorithm, but the run-time is longer than that of WIDIM-DPIV algorithm. Under the condition of strict real-time requirements, WIDIM-DPIV algorithm is preferred, while IM-WIDIM-DPIV algorithm is preferred under the condition of low real-time requirements.

## 4. Experiment

To study the practical effect of applying the IM-WIDIM-DPIV algorithm to the measurement of vertical upward oil–water two-phase immiscible flow, an experimental verification was conducted in

the oil–gas–water three-phase flow laboratory of the testing service branch of Daqing Oilfield Co., Ltd. in China. The experimental schematic diagram and experiment platform are shown in Figure 7.

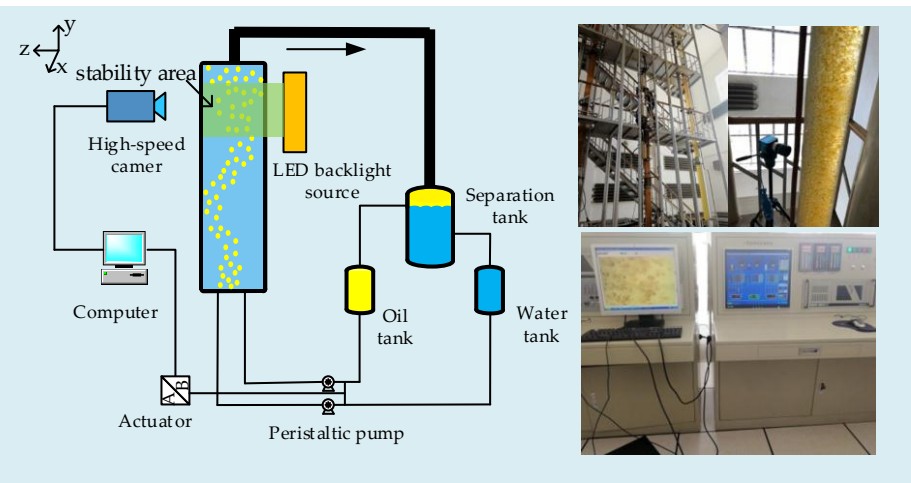

**Figure 7.** Experiment schematic diagram.

The area to be measured was illuminated with an LED backlight source. The led array lamps are designed by our team, and the parameters are as follows. (1) Light color: no-flicker white; (2) size: 20 cm × 20 cm, and evenly distributed with 19 × 4 LED lights; (3) input voltage: 220 V; (4) luminous flux: ~2400 lm. To make the lighting more uniform, the LED lights are assembled into a rectangular box without top lid, and cover the top of the box with an oil paper. The oil bubble in the vertical upward two-phase flow was used as the tracer in the experiment. A high-speed camera was adjusted to make flow velocity field imaging clear. As shown in Figure 7, Multiphase flow control system controls the flow velocity of oil phase and water phase according to the given velocity and holding rate. When the velocity table of oil phase and water phase has stable numerical value, the flow in pipe reaches the predetermined velocity and water up. The control errors of velocity and holding rate are less than 0.1%. After the oil and water phases are mixed and flowing about 6 m in the pipe, the flow state is steady. The high-speed camera is used to collect the image of oil–water two-phase immiscible flow, and then the DPIV image recorded by the high-speed camera was transmitted to a computer in real time for preservation and processing. Six iterations were carried out in the process of implementing the algorithm to calculate the flow velocity field, and the interrogation window was reduced to a quarter of the original size every two iterations. Interrogation window size changes from 128 pixel × 128 pixel to 32 pixel × 32 pixel. DPIV measurement experiment under different working conditions was conducted by changing different components of the multiphase flow through control system.

The flow velocity in this experiment ranged from $9.4 \times 10^{-3}$ m/s to $51.9 \times 10^{-3}$ m/s, with intergroup interval of $4.7 \times 10^{-3}$ m/s. The water cut ranged from 70% to 90%, with a step rate of 10%. The experimental parameters are of DPIV measurement experiment shown in Table 3.

**Table 3.** Experiment parameters.

| Parameter | Value |
|---|---|
| Dip angle | $0°$ |
| Pipe inner diameter | 125 mm |
| pipe outside diameter | 150 mm |
| object distance | 0.5 m |
| Length of steady flow section | 6.0 m |
| Fluid composition | tap water, diesel oil |
| Fluid Reynolds coefficient | 2638–6595 |
| Flow state | turbulent |
| Diesel density | $0.8 \text{ g/cm}^3$ |
| Light source | LED backlight source |
| Camera | HSVISION EoSens mini2 |
| Frame rate | 1000 frames/s |
| Image resolution | 1472 pixel × 1036 pixel |
| Measured area size | 180 mm × 144 mm |
| exposure time | 1 μs–1 s |
| Depth of focus | 0.8712 mm |
| Depth of field | 261 mm |
| Water phase velocity error | 0.10% |
| Oil phase velocity error | 0.10% |

As shown in Figure 8, there is optical distortion during the experiment as a result of the transparent curved surface of the vertical pipe. It has serious influence on flow velocity field measurement, the point whose original position is $l_o$ moves to $l_o'$ in the image due to the influence of optical distortion. It is necessary to revise the calculated flow velocity field as follows.

$$[v_r, u_l] = [v'_r, u'_l]\begin{bmatrix} 1 & 0 \\ 0 & \frac{n_1}{n_3 \cos(\delta_1 + \delta_2 - \tau_1 - \tau_2)} \end{bmatrix}, \tag{23}$$

$$\begin{cases} \delta_1 = \arcsin\left(\frac{l_o}{R_1}\right), \ \delta_2 = \arcsin\left(\frac{n_1 R_1 \sin \delta_1}{n_2 R_2}\right), \\ \tau_1 = \arcsin\left(\frac{n_1 \sin \delta_1}{n_2}\right), \ \tau_2 = \arcsin\left(\frac{n_1 R_1 \sin \delta_1}{n_3 R_2}\right), \end{cases} \tag{24}$$

where, $v'_r$ and $u'_l$ are the original longitudinal and transverse velocities at any point, respectively, $n_1$; $n_2$ and $n_3$ are the refractive indexes of liquid, glass, and air, respectively; $\delta_1$ and $\delta_2$ are the incidence angles; $\tau_1$ and $\tau_2$ are refraction angles; $l_o$ is the axial location of the point; and $R_1$ and $R_2$ are radii of the inside and outside pipes' diameter.

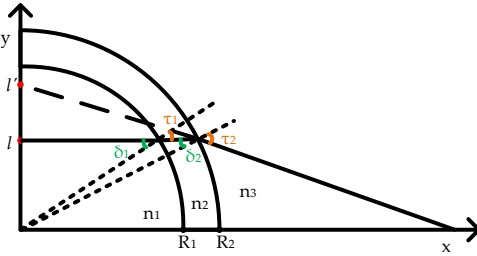

**Figure 8.** Optical distortion of the transparent curved surface of the vertical pipe.

## 5. Results and Discussion

### 5.1. Measurement Accuracy

Figure 9 shows the DPIV measurement results under the working condition of $51.9 \times 10^{-3}$ m/s-80% (flow velocity: $51.9 \times 10^{-3}$ m/s; water cut: 80%). To reduce the influence of image distortion by oil drops passing the light path, preprocess on the collected oil–water two-phase immiscible flow image is carried

out. Firstly, the regions of interest of images are cropped; then, a series of image processing techniques is adopted, such as Laplace sharpening and contrast enhancement to enhance useful information in the image. The experiment results show that the quality of the image is much better than that of the original image although there is still some distortion. Figure 9a,b show pairs of preprocessed images. The time interval is 0.006 s, and the measurement area size is 124 mm × 124 mm. Figure 9c,d shows the flow velocity fields measured by the WIDIM-DPIV algorithm and the IM-WIDIM-DPIV algorithm, respectively.

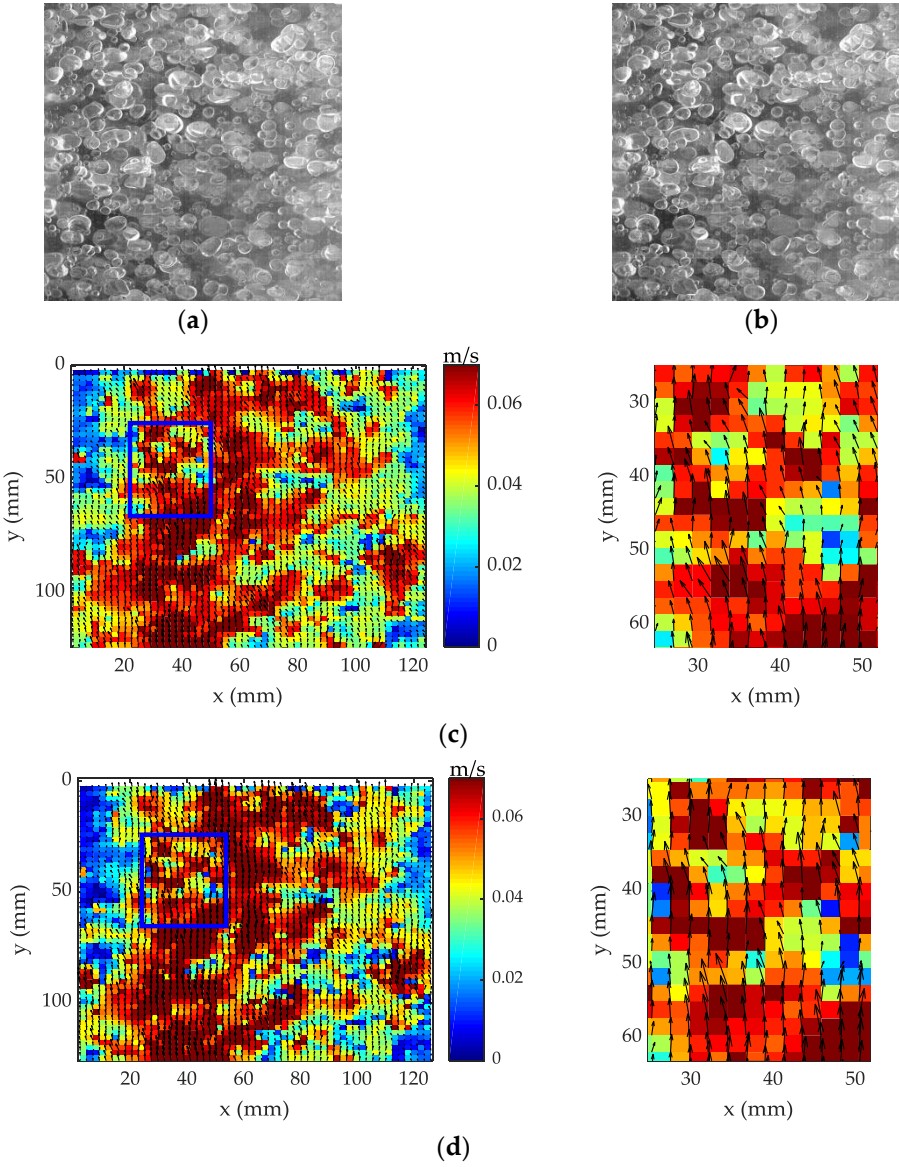

**Figure 9.** Measurement results at $51.9 \times 10^{-3}$ m/s-80%. (**a**,**b**) Preprocessed image pairs. (**c**) Flow velocity field by the WIDIM-DPIV algorithm. (**d**) Flow velocity field by the IM-WIDIM-DPIV algorithm.

According to the calculation, the average velocity of the WIDIM-DPIV algorithm was $47.12 \times 10^{-3}$ m/s and that of the IM-WIDIM-DPIV algorithm was $53.81 \times 10^{-3}$ m/s. The accuracy of the WIDIM-DPIV algorithm was 90.80% and that of the IM-WIDIM-DPIV algorithm was 96.31%. Therefore, the IM-WIDIM-DPIV algorithm improved the measurement accuracy.

To decrease the effect of experiment contingency on measurement accuracy, the DPIV experiment was conducted on the oil–water two-phase immiscible flow under different conditions in the inner diameter 125 mm vertical pipe. The flow velocity ranges from $9.4 \times 10^{-3}$ m/s to $51.9 \times 10^{-3}$ m/s with

intergroup interval of $4.7 \times 10^{-3}$ m/s. Figure 10 shows the measurement results at conditions of 90%, 80%, and 70% water cut.

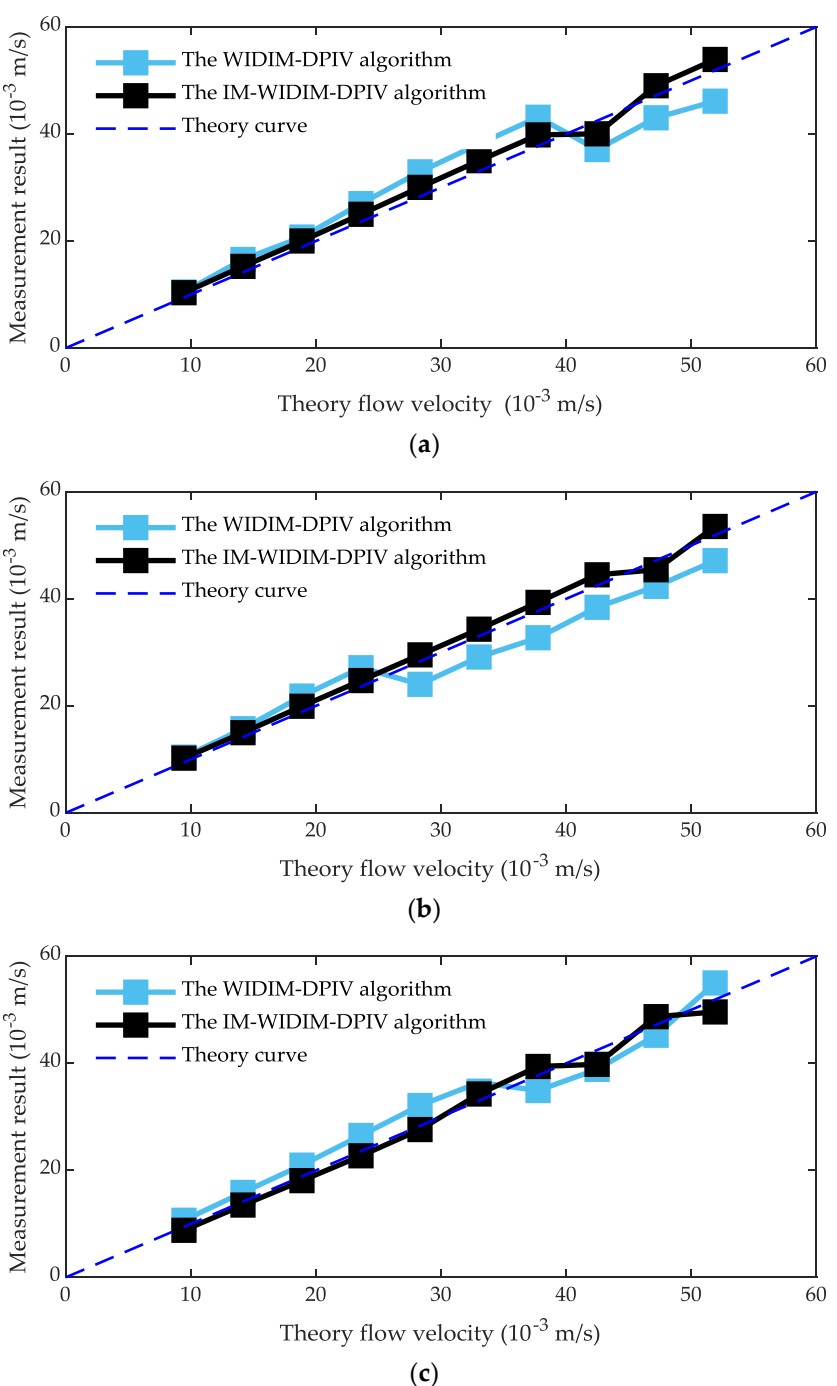

**Figure 10.** Measurement results under different water cut: (**a**) 90%, (**b**) 80%, and (**c**) 70%.

As shown in Figure 10, the measurement results of the IM-WIDIM-DPIV algorithm were closer to the theoretical values for all water cut than those of WIDIM-DPIV algorithm. This qualitatively indicates that the accuracy of the IM-WIDIM-DPIV algorithm was higher than that of the WIDIM-DPIV algorithm. Under different working conditions, the mean error of the WIDIM-DPIV algorithm was $3.39 \times 10^{-3}$ m/s and that of the IM-WIDIM-DPIV algorithm was $1.43 \times 10^{-3}$ m/s. This quantitatively indicates that the mean error of the IM-WIDIM-DPIV algorithm was smaller than that of WIDIM-DPIV algorithm and the measurement results of the IM-WIDIM-DPIV algorithm were closer to the theory

value under all working conditions. According to all measurement data, the average accuracy of the IM-WIDIM-DPIV algorithm was 94.8%. It was higher than that of the WIDIM-DPIV algorithm, which was 88.3%. The standard deviation of the WIDIM-DPIV algorithm was $1.39 \times 10^{-3}$ m/s and that of the IM-WIDIM-DPIV algorithm was $0.56 \times 10^{-3}$ m/s, which indicates that the volatility of the IM-WIDIM-DPIV algorithm was smaller than that of WIDIM-DPIV algorithm. The maximum deviation of the WIDIM-DPIV algorithm was $5.77 \times 10^{-3}$ m/s under the condition of $51.9 \times 10^{-3}$ m/s-90% and that of the IM-WIDIM-DPIV algorithm was $2.65 \times 10^{-3}$ m/s under the condition of $42.4 \times 10^{-3}$ m/s-70%. The maximum relative deviation of the WIDIM-DPIV algorithm was 16.8% under the condition of $14.1 \times 10^{-3}$ m/s-90%, and that of the IM-WIDIM-DPIV algorithm was 9.4% under the condition of $9.4 \times 10^{-3}$ m/s-90%, which indicates that the deviation degree of result that calculated by the IM-WIDIM-DPIV algorithm from the actual value was lower than that of the WIDIM-DPIV algorithm.

As shown in Figure 10, there is jumping data at specific points with different water cut, and the major differences between the two PIV approaches occur at relatively high flow velocity. This may be caused by the following four reasons. (1) Oil droplets are used as tracer particles in the experiment. With the increase of flow velocity and water cut, the average size of oil droplets increase due to the stacking effect. The minimum visible diameter of oil droplets is ~2 mm, the maximum diameter is ~9 mm, and the average diameter is ~4.5 mm. In the case of the same flow velocity and different water cut, the oil droplets distribution in shooting area is different. When the water cut is high, the oil droplets distribution in the shooting area is sparse compared with that of low water cut, and the number of matched oil droplets is small. This results in larger errors. (2) Reversal flows induced by buoyancy driven turbulence. (3) With the increase of flow velocity, the velocity gradient in the pipe increases, and DPIV interrogation window's following performance becomes worse. However, the interrogation window following performance of IM-WIDIM-DPIV algorithm is better than that of WIDIM-DPIV algorithm. Therefore, the measurement accuracy difference is obvious at high flow velocity. (4) As the flow velocity and water cut increases, the distribution of oil droplets gradually became denser; when the droplets are too dense, oil droplets shield each other seriously. This results in inaccurate calculation in DPIV calculation, so the accuracy of DPIV reduces.

In conclusion, from both qualitative and quantitative perspectives, the IM-WIDIM-DPIV algorithm had a higher precision than the WIDIM-DPIV algorithm, with a maximum measurement accuracy of 97.5% and an average measurement accuracy of 94.8%.

### 5.2. Measurement Reproducibility

To verify the stability of the IM-WIDIM-DPIV algorithm, the oil–water two-phase immiscible flow in the inner diameter 125 mm vertical upward pipe was repeatedly measured under different water cut and flow velocity of $9.4 \times 10^{-3}$ m/s - $51.9 \times 10^{-3}$ m/s with intergroup interval of $4.7 \times 10^{-3}$ m/s. The measurement results are shown in Figure 11 and Table 4.

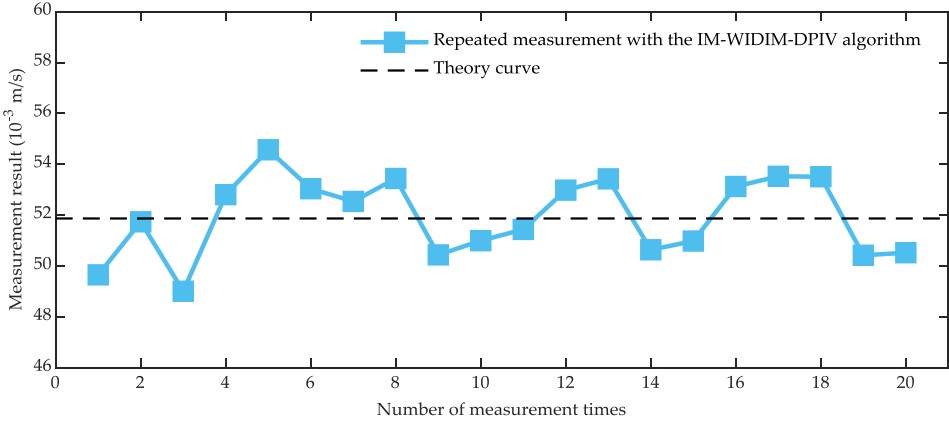

**Figure 11.** Repeated measurement results at $51.9 \times 10^{-3}$ m/s-80%.

As shown in Figure 11, the measured curve is effectively consistent with the theory curve, despite fluctuating slightly. The calculated maximum fluctuation was $2.93 \times 10^{-3}$ m/s, with a relative deviation of 5.64%, standard deviation of $1.50 \times 10^{-3}$ m/s, and average value of $51.74 \times 10^{-3}$ m/s. This is close to the theoretical value of $51.90 \times 10^{-3}$ m/s, so the measured results have good stability.

**Table 4.** Repeated velocity measurement results.

| Flow Velocity ($10^{-3}$ m/s) | Water Cut 90% | | | Water Cut 80% | | | Water Cut 70% | | |
| --- | --- | --- | --- | --- | --- | --- | --- | --- | --- |
| | Max-Deviation ($10^{-3}$ m/s) | Repeatability (%) | | Max-Deviation ($10^{-3}$ m/s) | Repeatability (%) | | Max-Deviation ($10^{-3}$ m/s) | Repeatability (%) | |
| 9.4 | 4.64 | 1.51 | 2.79 | 3.02 | 3.87 | 1.89 |
| 14.1 | 2.19 | 1.03 | 5.72 | 2.78 | 7.36 | 1.26 |
| 18.9 | 8.50 | 5.81 | 8.44 | 2.98 | 8.13 | 1.91 |
| 23.6 | 5.35 | 5.07 | 6.14 | 1.15 | 4.27 | 1.32 |
| 28.3 | 10.13 | 4.34 | 4.61 | 0.91 | 4.33 | 0.83 |
| 33.0 | 3.63 | 2.77 | 5.13 | 1.67 | 6.53 | 0.74 |
| 37.7 | 9.54 | 2.61 | 5.49 | 0.97 | 2.12 | 0.88 |
| 42.4 | 7.42 | 1.65 | 3.35 | 1.94 | 3.24 | 1.04 |
| 47.2 | 9.48 | 1.57 | 4.76 | 1.19 | 6.24 | 1.93 |
| 51.9 | 5.86 | 1.08 | 2.22 | 2.93 | 2.44 | 0.76 |

Table 4 shows that the overall average maximum deviation of repeated measurements under various working conditions was $5.46 \times 10^{-3}$ m/s; the highest was $10.13 \times 10^{-3}$ m/s and the lowest was $2.12 \times 10^{-3}$ m/s. The average reproducibility was 1.98%; the highest was 5.81% and the lowest was 0.74%, all of which did not exceed 10%. When the water cut was 70%, the average maximum deviation and average reproducibility were the minimum. They were $4.85 \times 10^{-3}$ m/s and 1.26%, respectively. Therefore, the IM-WIDIM-DPIV algorithm has good reproducibility and stability.

Figure 12 shows the velocity distribution of oil–water two-phase immiscible flow in the inner diameter 125 mm vertical upward pipe under different working conditions obtained by the IM-WIDIM-DPIV algorithm.

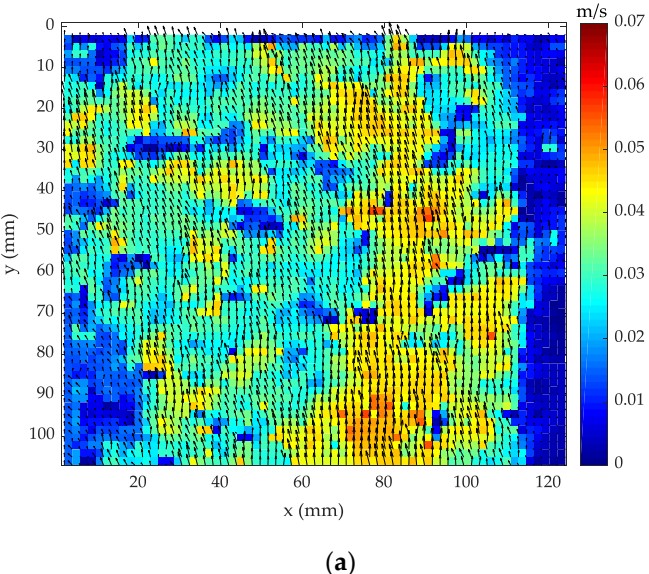

(**a**)

**Figure 12.** *Cont*.

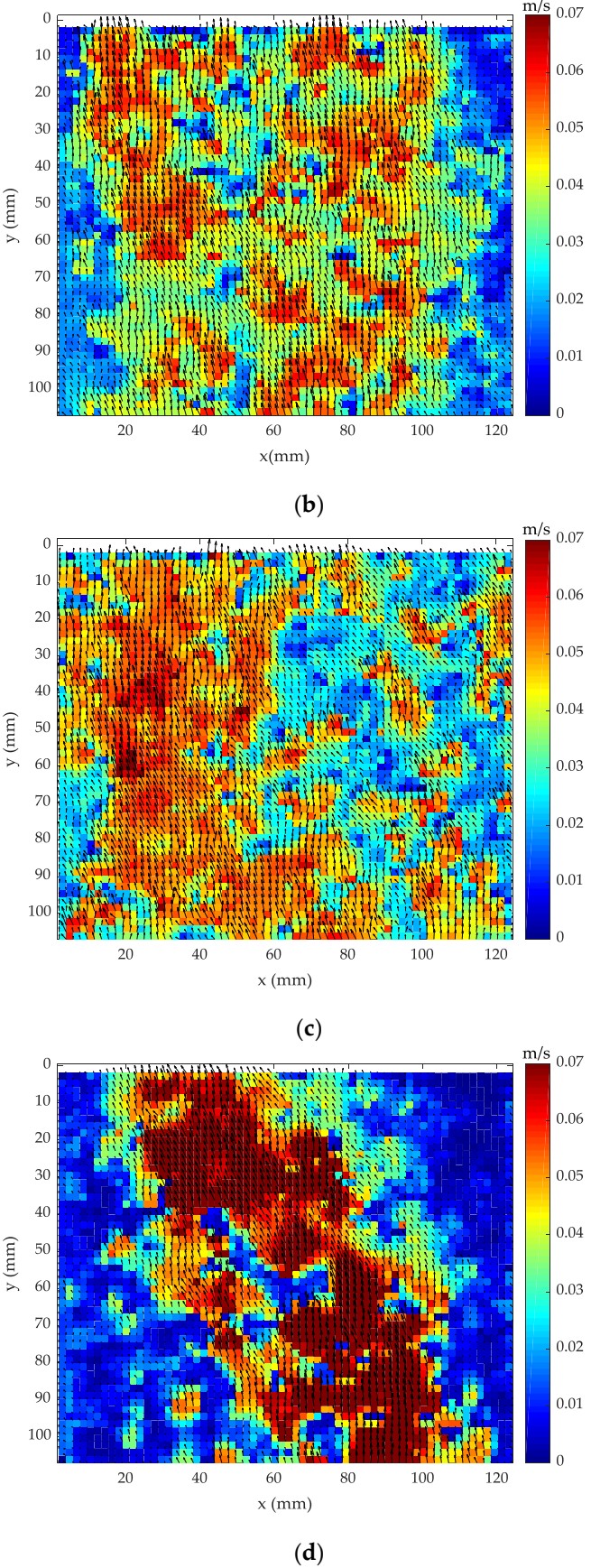

(**b**)

(**c**)

(**d**)

**Figure 12.** *Cont.*

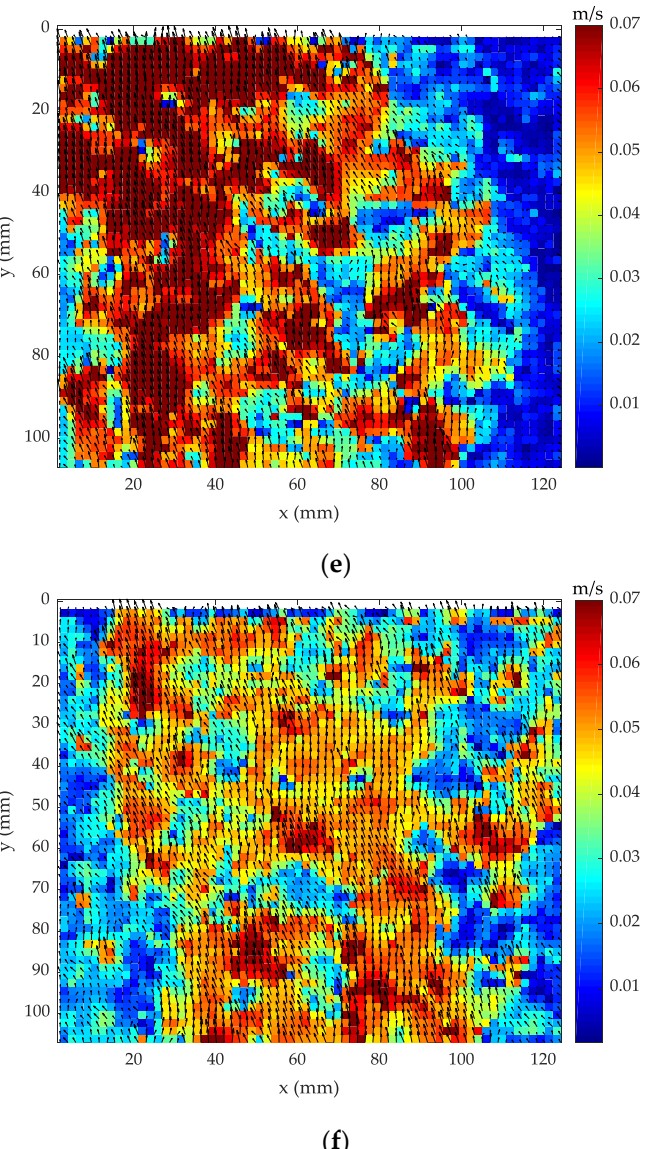

(e)

(f)

**Figure 12.** Measurement results under different working conditions. (**a**) Flow velocity field of $28.3 \times 10^{-3}$ m/s-70%. (**b**) Flow velocity field of $37.7 \times 10^{-3}$ m/s-70%. (**c**) Flow velocity field of $51.9 \times 10^{-3}$ m/s-70%. (**d**) Flow velocity field of $47.2 \times 10^{-3}$ m/s-90%. (**e**) Flow velocity field of $47.2 \times 10^{-3}$ m/s-80%. (**f**) Flow velocity field of $47.2 \times 10^{-3}$ m/s-70%.

Figure 12 shows that the axial velocity of the flow in the pipe is clearly greater than the radial velocity, so the fluid flows upward as a whole, and the average velocity at the center of the vertical pipe is significantly higher than that near the boundary; this is consistent with the actual bubble flow in vertical wells [40,41].

*5.3. Measurement Uncertainty Evaluation*

The velocity calculation equation based on DPIV is Equation (25):

$$|v| = \gamma \frac{\Delta piexl}{\Delta t},\qquad(25)$$

where, $\gamma$ is camera calibration factor, $\Delta pixel$ is the pixel value of particles movement in two adjacent frames of images, and $\Delta t$ is the time difference between two frames of images.

To evaluate the uncertainty measurement of oil–water two-phase immiscible flow velocity, the above equation is equivalent to

$$|v| = \frac{4(\Delta Q_W + \Delta Q_O)}{\pi D^2 \Delta t}, \tag{26}$$

where, $\Delta Q_W$ is the water phase flow rate, $\Delta Q_O$ is the oil phase flow rate, and $D$ is pipe inner diameter.

In the calculation process, the measurement of oil–water two-phase immiscible flow velocity is obtained according to Equation (25), not Equation (26). According to Equation (25), flow velocity measurement errors are divided into two categories: One is instrument calibration error, which is mainly composed of object distance error, camera tilt error, and camera manufacturing error; the other is the geometric error and control error of measurement system.

According to the above analysis, the measurement uncertainty of oil–water two-phase immiscible flow measurement mainly includes the following.

(1) Measurement uncertainty $u_w$ caused by the control error of water phase flow velocity $\Delta Q_W$.

The maximum value of this error is 0.1%, the error approximately follows uniform distribution, and the maximum flow velocity of water phase is 3.20 m³/h. Therefore, the measurement uncertainty caused by this part is calculated as

$$u_w = \left(0.072 \times 10^{-3}\right)/\sqrt{3}\ \text{m/s}, \tag{27}$$

(2) Measurement uncertainty $u_o$ caused by the control error of water phase flow rate $\Delta Q_O$

The maximum value of this error is 0.1%, the error approximately follows uniform distribution, and the maximum flow velocity of oil phase is 1.38 m³/h. Therefore, the measurement uncertainty caused by this part is calculated as

$$u_o = \left(0.032 \times 10^{-3}\right)/\sqrt{3}\ \text{m/s}, \tag{28}$$

(3) Measurement uncertainty $u_D$ caused by the error of pipe inner diameter $D$

The maximum value of this error is 0.1 mm, pipe inner diameter $D$ is 125 mm, and the error approximately follows uniform distribution. Therefore, the measurement uncertainty caused by this part is

$$u_D = \left(0.003 \times 10^{-3}\right)/\sqrt{3}\ \text{m/s}, \tag{29}$$

(4) Measurement uncertainty $u_{\Delta t}$ caused by the error of time difference $\Delta t$ of per camera frame

The maximum value of this error is 0.05%, the error approximately follows uniform distribution, and the time difference of each frame is 0.001. Therefore, the measurement uncertainty caused by this part is

$$u_D = \left(0.002 \times 10^{-3}\right)/\sqrt{3}\ \text{m/s}, \tag{30}$$

(5) Measurement uncertainty $u_\gamma$ caused by the error of camera calibration $\gamma$

Object distance error is 1 mm, object distance is 0.5 m, the tilt angle error of the camera is 0.01°, the camera design angle is 0°, and the errors are within the limits permitted of the measurement procedure. Therefore, the measurement uncertainty of this part according to the manufacturer is

$$u_\gamma = (0.05\% + \text{L} \times 0.005\%)V_{\text{max}} = 0.059 \times 10^{-3}\ \text{m/s} \tag{31}$$

where, $L$ is object distance, $V$max is the max flow velocity, and the value is 0.113 m/s.

Therefore, the combined uncertainty $U_{95}$ of the oil–water two-phase immiscible flow measurement system is

$$U_{95} = 2\sqrt{u_w^2 + u_o^2 + u_D^2 + u_{\Delta t}^2 + u_\gamma^2} = 0.149 \times 10^{-3}\ \text{m/s} \tag{32}$$

## 6. Conclusions

The reasons for the low DPIV image matching rate were studied in this paper. It was found that the image noise and the poor window following performance results the poor deformation performance of the interrogation window. To improve the deformation performance of the interrogation window, and thus improve the accuracy of the algorithm, ICP and MLS are introduced into WIDIM algorithm in the DPIV postprocessing algorithm. The improved DPIV algorithm is called the IM-WIDIM-DPIV algorithm. The measuring accuracy of the IM-WIDIM-DPIV algorithm is higher than that of WIDIM-DPIV and the uncertainty of measurement is $0.149 \times 10^{-3}$ m/s. The reproducibility of the experimental data is 1.98%. This provides a feasible method for accurately drawing the oil–water two-phase immiscible flow velocity field. According to the flow velocity filed, optimization design of the logging tool is carried out. For example, turbine flowmeter is optimized according to the flow velocity filed of different flow conditions.

**Author Contributions:** Methodology, L.H. and Y.C.; Software, X.L. and L.H.; Validation, C.F., Y.C., and X.L.; writing—original draft preparation, L.H. and Y.C.; Writing—review and editing, L.H., Y.C., and C.F.; Funding acquisition, C.F.

**Funding:** This research was funded by the National Science Foundation of China (51774092) and the China Postdoctoral Science Foundation (2016M601399).

**Acknowledgments:** The authors thank Dai Yu (Nankai University, Tianjin, China) for his advice on mathematical program used in this work.

**Conflicts of Interest:** The authors declare no conflict of interest.

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
