# Peer review of "Flow Velocity Field Measurement of Vertical Upward Oil–Water Two-Phase Immiscible Flow Using the Improved DPIV Algorithm Based on ICP and MLS"

_applsci, doi:10.3390/app9163292_

Round 1

Reviewer 1 Report

Lianfu et al. report about the “Flow velocity field measurement of vertical upward 2 oil-water two-phase flow based on the improved 3 digital particle image velocimetry algorithm”. The authors applied PIV measurement and various data evaluation methods to refine the flow analysis in a two-phase flow of a water-oil mixture. The presented approach sounds interesting and might be publishable after revision.

More information of the experimental setup is necessary:

·         Within the text, the authors claim a background (which I understood as volume) illumination instead of a light sheet, how could they achieve planar velocity fields, especially with only one camera? Contrarily, in figure 1 b a light-sheet is presented. Please explain and change the text if necessary. In my opinion there was a light-sheet applied, otherwise the complete study would be corrupt. By the way, did the authors really apply a cylindrical mirror instead of a cylindrical lens (see figure 1)?

·         From the sketch in figure 1 it is not is not clear which plane of the velocity field was measured, how and why. What is the x-y-plane? It would be useful to put a coordinate system to the sketch in figure 1 and 7. In figure 1, it seems that the light-sheet is diagonally oriented to the pipe.

·         What was the depth of focus and how to eliminate image distortion by oil drops passing the light path?

·         What is the measured/evaluated area size, especially compared to the diameter of the pipe?

·         Finally, all this information has to be explicitly written in the experimental section and not only to be shown in images. Otherwise the understanding is very diffuse to the reader.

·         Is the flow laminar or turbulent? What is the Reynolds number?

For the sake of completeness, what is the reason for the jumping data at specific points for different water cuts with standard algorithm in figure 10, even though the improved algorithm corrects this distortion? What about reversal flows induced by buoyancy driven turbulence (the density of oil drops is smaller than that of water)? May this be the reason? Furthermore, would then the original approach be the better one, since the improved algorithm removes the reversals?

Further points

·         Figure 6 and Figure 9: The reader cannot realize any qualitative or quantitative difference between the images of the different analysis methods. The authors should additionally provide difference images between both methods to emphasize the deviations.

·         Figure 7: Sepration tank à Separation tank

·         Figure 9: the scaling numbers of the colorbars are cut-off; units are missing (length scales in cm or mm to have a comparison to the dimension of the pipe)

·         Figure 12: units are missing (see above)

If the experimental procedure cannot be addressed adequately, the manuscript cannot be published.

Author Response

Dear Ms. Allen Dou and Reviewers,

   Thank you for your letter and for the reviewers’ comments concerning our manuscript. Those comments are very helpful for revising and improving our paper.

We have made correction which we hope meet with approval. Revised portions are marked by underlining in the paper.

Special thanks to you for your good comments.

We hope that these revisions are satisfactory and that the revised version will be acceptable for publication in the journal Applied Sciences. Thank you very much for all your help and looking

forward to hearing from you soon.
Wish you all the best!

Sincerely yours,
Changfeng Fu

Reviewer 2 Report

The opening sections are introduced poorly. It is difficult to understand why the current software is in used that has to be improved. Moreover, the information regarding the current equipment and resolution used in the experiments are not provided. Moreover, the uncertainty in the data is not reported. The conclusions provided is poorly supported. Based on few experiments and cases studied in this article, they cannot comment on the accuracy of the data or method.

Author Response

(The authors gave the same response as above.)

Reviewer 3 Report

Please see my comments in attachment.

Author Response

(The authors gave the same response as above.)

Round 2

Reviewer 1 Report

Dear authors,

very nice and proper revision. The manuscript can be published as is.

Please carefully check again for grammar and orthography during proof corrections.

best regards

Author Response

Dear Ms. Allen Dou and Reviewers,

   Thank you for your letter and for the reviewers’ comments concerning our manuscript. Those comments are very helpful for revising and improving our paper.

We have made correction which we hope meet with approval. Revised portions are marked by underlining in the paper. The manuscript had been corrected by a native speaker. We have checked carefully the English language with the help of a foreign students from Australia.

Special thanks to you for your good comments.

We hope that these revisions are satisfactory and that the revised version will be acceptable for publication in the journal “Applied Sciences”. Thank you very much for all your help and looking forward to hearing from you soon.

Wish you all the best!

Sincerely yours,

Changfeng Fu

Reviewer 3 Report

Please see in attachment.

Author Response

Dear  Reviewers,

   Thank you for your letter and for the reviewers’ comments concerning our manuscript. Those comments are very helpful for revising and improving our paper.

We have made correction which we hope meet with approval. Revised portions are marked by underlining in the paper. The manuscript had been corrected by a native speaker. We have checked carefully the English language with the help of a foreign students from Australia.

Special thanks to you for your good comments.

We hope that these revisions are satisfactory and that the revised version will be acceptable for publication in the journal “Applied Sciences”. Thank you very much for all your help and looking forward to hearing from you soon.

Wish you all the best!

Sincerely yours,

Changfeng Fu
